# Cancer Cell Secreted Legumain Promotes Gastric Cancer Resistance to Anti-PD-1 Immunotherapy by Enhancing Macrophage M2 Polarization

**DOI:** 10.3390/ph17070951

**Published:** 2024-07-16

**Authors:** Xu Pei, Shi-Long Zhang, Bai-Quan Qiu, Peng-Fei Zhang, Tian-Shu Liu, Yan Wang

**Affiliations:** 1Department of Medical Oncology, Zhongshan Hospital, Fudan University, Shanghai 200032, China; 21111220108@m.fudan.edu.cn (X.P.); zhang.shilong@zs-hospital.sh.cn (S.-L.Z.); zhang.pengfei@zs-hospital.sh.cn (P.-F.Z.); 2Department of Cardiothoracic Surgery, The Second Affiliated Hospital of Nanchang University, Nanchang 330030, China; qiubaiquan@163.com; 3Department of Medical Oncology, Shanghai Geriatric Medical Center, Shanghai 201104, China; 4Cancer Center, Zhongshan Hospital Affiliated to Fudan University, Shanghai 200032, China; 5Center of Evidence-Based Medicine, Zhongshan Hospital Affiliated to Fudan University, Shanghai 200032, China; 6Shanghai Medical College, Zhongshan Hospital Immunotherapy Translational Research Center, Shanghai 200032, China

**Keywords:** gastric cancer, legumain, immunotherapy, macrophages

## Abstract

The interaction between cancer cells and immune cells plays critical roles in gastric cancer (GC) progression and immune evasion. Forced legumain (LGMN) is one of the characteristics correlated with poor prognosis in gastric cancer patients. However, the role of gastric-cancer-secreted LGMN (sLGMN) in modulating the tumor immune microenvironment and the biological effect on the immune evasion of gastric cancer remains unclear. In this study, we found that forced expression of sLGMN in gastric cancer serum correlates with increased M2 macrophage infiltration in GC tissues and predicted resistance to anti-PD-1 immunotherapy. Mechanistically, gastric cancer cells secrete LGMN via binding to cell surface Integrin αvβ3, then activate Integrin αvβ3/PI3K (Phosphatidylinositol-4,5-bisphosphate3-kinase)/AKT (serine/threonine kinase)/mTORC2 (mammalian target of rapamycin complex 2) signaling, promote metabolic reprogramming, and polarize macrophages from the M1 to the M2 phenotype. Either blocking LGMN, Integrin αv, or knocking out Integrin αv expression and abolishing the LGMN/Integrin αvβ3 interaction significantly inhibits metabolic reprogramming and polarizes macrophages from the M1 to the M2 phenotype. This study reveals a critical molecular crosstalk between gastric cancer cells and macrophages through the sLGMN/Integrinαvβ3/PI3K/AKT/mTORC2 axis in promoting gastric cancer immune evasion and resistance to anti-PD-1 immunotherapy, indicating that the sLGMN/Integrinαvβ3/PI3K/AKT/mTORC2 axis may act as a promising therapeutic target.

## 1. Introduction

Gastric cancer (GC) is one of the malignant tumors with the highest incidence and mortality rates in the world [1]. Recently, immune checkpoint inhibitors (ICIs) such as PD-1 (Programmed cell death 1)/PD-L1 (Programmed cell death 1 ligand 1) monoclonal antibodies (mAbs) have partially improved patient prognosis; however, some patients have shown primary or acquired resistance to PD-1/PD-L1 mAbs [2,3,4]. The mechanism of immune evasion in gastric cancer is complex, involving not only changes in tumor cells themselves but also adaptive changes in various cells in the tumor immune microenvironment, such as tumor-associated fibroblasts, macrophages, cytotoxic T lymphocytes, and dendritic cells [5,6,7]. Recent studies have reported that activation of tumor immune evasion pathways is one of the critical mechanisms for resistance to PD-1/PD-L1 mAbs [8]. Therefore, it is of great significance to thoroughly explore the molecular mechanisms of immune evasion in GC and predict the efficacy of immunotherapy.

Legumain (LGMN), also named asparaginyl endopeptidase, is one of the members of the C13 family of cysteine proteases. LGMN exerts its biological functions by specifically cleaving peptide bonds with asparaginyl residues [9]. LGMN can be expressed on the cell membrane, intracellularly, and secreted into the extracellular space and enter the peripheral blood [10,11]. The expression of LGMN is elevated in various tumors and tumor-associated macrophages (TAMs). In addition, forced LGMN expression was associated with poorer prognosis in several cancers [12,13]. For instance, the interaction between LGMN and Integrin αvβ3 significantly promotes tumor cell migration and invasion [10]. TAMs are an integral part of the tumor microenvironment and belong to the M2-type macrophages [14]. TAMs promote tumor stroma remodeling, neovascularization, tumor cell proliferation, and metastasis by secreting various growth factors, angiogenic factors, and proteases [14,15]. Recent studies have reported that knocking out LGMN expression in TAMs, thereby generating tumor-bearing mice with LGMN deficiency, significantly inhibits tumor growth. Analysis of primary tumor cells indicates that knocking out LGMN expression in TAMs inhibits cancer cell proliferation and induces cancer cell apoptosis [16]. Loss of LGMN expression in TAMs can significantly accelerate tumor cell senescence in vivo [17]. Elevated expression of LGMN in TAMs within GC tissues, and knocking down LGMN expression in TAMs, can effectively suppress tumor growth [18].

The molecular mechanisms underlying sLGMN-mediated intercellular communication between gastric cancer cells and TAMs remain unclear. Here, we employed co-culture, co-immunoprecipitation, and flow cytometry to investigate how tumor-derived sLGMN influences the phenotype and function of TAMs. In conclusion, our results demonstrated that gastric cancer exhibits a higher expression of LGMN and secretes LGMN extracellularly. Subsequently, sLGMN binds to Integrin αv on M2 macrophages, leading to the upregulation of the mTORC2 (mammalian target of rapamycin complex 2) pathway activity. This process facilitates the polarization of M1 macrophages towards the M2 phenotype, contributing to resistance PD-1 mAbs. These findings imply that combination therapy involving mTORC2 inhibitors may offer a promising method for counteracting immune evasion in gastric cancer and augmenting the effectiveness of PD-1 mAbs. This study significantly contributes to elucidating the precise molecular mechanisms that cancer-cell-secreted LGMN (sLGMN) modulates metabolic reprogramming in M1 macrophages, thus promoting their polarization towards the M2 phenotype and facilitating immune evasion in GC. These insights lay the groundwork for reversing immune evasion in GC and identifying novel immunotherapy targets.

## 2. Results

### 2.1. Forced Serum LGMN Expression Is Associated with Increased M2 Macrophages in GC Tissues

Our previous research indicates that LGMN is highly expressed in gastric cancer and correlates with poor prognosis [19]; however, the biological functions of LGMN in gastric cancer are largely unknown. In order to elucidate whether the high expression of LGMN is associated with immune evasion in gastric cancer, we conducted an analysis of TCGA (The Cancer Genome Atlas) database, and the results indicated that LGMN expression was associated with increased M2 macrophage infiltration in gastric cancer tissues (Figure 1A). These results suggest that LGMN overexpression might serve as a promising molecular marker for predicting the efficacy of immune therapeutic interventions in GC. Then, we randomly selected 20 patients with stage IV gastric cancer treated with anti-PD-1 therapy in our hospital (all of the patients were pMMR (mismatch repair-proficient), TPS (tumor proportion score) ≥ 5%) for retrospective analysis. Among them, there was 1 patient with PR (partial remission), 5 patients with SD (stable disease), and 14 patients with PD (disease progression). The serum levels of LGMN in the 20 patients were detected by ELISA (enzyme-linked immunosorbent assay), and the patients were divided into LGMN^high^ and LGMN^low^ groups according to the median value. Subsequently, we defined PR and SD patients as the therapeutic sensitive group and PD patients as the therapeutic resistance group. The results showed that the expression level of serum LGMN in the therapeutic resistance group was significantly higher than that in the sensitive group (Figure 1B). Furthermore, the macrophage infiltration in tumor tissues was detected by immunohistochemistry. The results showed the LGMN^high^ group tumor tissues with increased infiltration of M2 macrophages and decreased infiltration of M1 macrophages compared to the LGMN^low^ group (Figure 1C,D). Taken together, the above results suggest that LGMN overexpression in gastric cancer may induce M2 polarization of macrophages, and then induce immune escape and resistance to anti-PD-1 therapy.

### 2.2. LGMN Promotes M1 Macrophages Polarization towards M2 Phenotype 

In order to further clarify the biological function of secreted LGMN (sLGMN) in gastric cancer immune evasion, the levels of LGMN in seven GC cell lines (HCG27, BGC823, MKN45, MKN28, SGC7901, AGS, MGC803) were examined by Western blot. The results showed that LGMN expression was relatively low in AGS and BGC823 cells, while the expression was relatively higher in HGC27 and MGC803 cells (Appendix A). In addition, ELISA was used to detect the levels of LGMN in the culture supernatant of these seven cell lines. The results showed that the levels of LGMN in the culture supernatant were consistent with those in parental cells (Appendix A). Then, LGMN overexpressing (ASG and BGC823) GC cells were established (ASG-L and BGC823-L). The results demonstrated that the secretion of LGMN increased in AGS and BGC823 cells with increased LGNM expression (Appendix A). Co-culture of AGS-L and BGC823-L cells with THP-1-derived M1 macrophages was performed. In addition, flow cytometry was employed to examine whether THP-1-derived M1 macrophages polarized towards the M2 subtype. The results indicated that AGS-L and BGC823-L cells significantly promoted the polarization of THP-1-derived M1 macrophages towards the M2 subtype (as evidenced by a significant increase in the expression levels of CD163 and CD206) (Figure 2A). Additionally, the significant promotion of M1 to M2 polarization by AGS-L and BGC823-L cells could be blocked by neutralizing antibodies against LGMN (Figure 2A). To further exclude interference from other factors in the co-culture system, we stimulated THP-1-derived M1 macrophages with recombinant human LGMN (rh-LGMN) and assessed their polarization towards the M2 phenotype using flow cytometry. The results showed that rh-LGMN significantly promoted the polarization of THP-1-derived M1 macrophages towards the M2 phenotype (Figure 2B). Subsequently, we induced differentiation of PBMCs (peripheral blood monocytes) from healthy individuals into M1 macrophages using IFN-γ (interferon-γ) (Figure 2C). Then, the M1 macrophages derived from healthy individuals’ monocytes were cultured with rh-LGMN for 5 days. Flow cytometry results indicated that rh-LGMN promoted polarization of monocytes-derived M1 macrophages towards the M2 phenotype (Figure 2D).

### 2.3. LGMN Upregulates Integrin αvβ3/AKT (Serine/Threonine Kinase)/mTORC2 Axis Signaling in Macrophages

Previous research has reported that tumor cells promote tumor cell metastasis by secreting LGMN, which further binds to Integrin αvβ3 on the tumor cell membrane [10]. Therefore, we hypothesize that sLGMN may act as a ligand for Integrin αv or β3, activating integrin-related signaling pathways, thereby upregulating mTORC2 signaling pathway activity, promoting metabolic reprogramming of M1 macrophages, and polarizing them towards M2 macrophages. Then, by stimulating THP-1-derived M1 macrophages with recombinant LGMN protein, the results showed that rh-LGMN significantly upregulated the mTORC2 signaling pathway activity of THP-1-derived M1 macrophages (Figure 3A). Additionally, rh-LGMN-induced polarization of THP-1-derived M1 macrophages towards M2 could be blocked by an Integrin αv antibody but not by an Integrin β3 antibody (Figure 3B). Furthermore, knocking down the expression of Integrin β3 in THP-1-derived M1 macrophages via lentivirus-mediated shRNA inhibited rh-LGMN-induced M2 polarization of macrophages (Figure 3C,D). These results demonstrate that LGMN may promote M1 macrophage polarization towards M2 by binding to Integrin αv, and the formation of a complex with Integrin β3 is necessary for this process. Furthermore, rh-LGMN significantly upregulated the mTORC2 signaling pathway activity of peripheral blood monocytes-derived M1 macrophages from healthy individuals (Figure 3E), accompanied by upregulation of CD36 expression (Figure 3F).

### 2.4. mTOR (Mammalian Target of Rapamycin) Inhibitors Block LGMN-Induced Polarization of M1 Macrophages towards M2

To further confirm whether LGMN induces the polarization of M1 macrophages towards M2 via upregulation of the Integrin αvβ3/AKT/mTORC2 axis signaling pathway activity, we conducted flow cytometry to determine whether the mTORC1/2 inhibitor AZD2014 could block rh-LGMN-mediated polarization of M1 macrophages (derived from healthy donors’ monocytes) towards M2 phenotype. The results showed that AZD2014 significantly inhibited rh-LGMN-mediated polarization of M1 macrophages towards the M2 phenotype (Figure 4A). Western blot results indicated that AZD2014 significantly inhibited the upregulation of mTORC2 signaling pathway activity in LGMN-mediated M1 macrophages (Figure 4B). Importantly, the mTORC1 signaling pathway activity in LGMN-mediated M1 macrophages was not significantly affected (Figure 4C). The above study suggests that rh-LGMN primarily regulates the mTORC2-associated signaling pathway, rather than the mTORC1-associated signaling pathway in M1 macrophages.

### 2.5. LGMN Induces Metabolic Reprogramming of M1 Macrophages via Integrin αvβ3/AKT/mTORC2 axis

Previous studies have reported that enhanced activity of AKT/mTORC2 signaling pathway-mediated metabolic reprogramming plays an important role in the polarization of macrophages to the M2 phenotype. The ECAR (Extracellular Acidification Rate) experiment results indicate that rh-LGMN induces metabolic reprogramming in M1 macrophages derived from peripheral blood monocytes of healthy donors, primarily characterized by increased glycolysis (Figure 5A). By culturing cells from each group in DMEM (Dulbecco’s Modified Eagle Medium) containing [9, 10(n)-3H] oleic acid for 12 h, followed by extraction of the aqueous phase containing 3H_2_O from the chloroform/methanol (2:1 *v*/*v*) supernatant, the level of fatty acid oxidation (FAO) in each group of cells was measured. The experimental results suggest that rh-LGMN can promote lipid metabolism in M1 macrophages (Figure 5B). OCR (oxygen consumption rate) experiments indicate that rh-LGMN significantly upregulates oxidative phosphorylation levels and fatty acid uptake in M1 macrophages derived from PBMCs of healthy donors (Figure 5C,D). Additionally, the mTORC1/2 inhibitor AZD2014 was found to inhibit LGMN-induced metabolic reprogramming in M1 macrophages (Figure 5E,F). Given that the above research results suggest that rh-LGMN upregulates AKT/mTORC2 activity by binding to Integrin αv mechanisms, we speculate that Integrin αv-blocking antibodies may inhibit LGMN-induced metabolic changes in M1 macrophages. We stimulated M1 macrophages derived from healthy donors with rh-LGMN and added Integrin αv or β3-blocking antibodies separately, and the ECAR and OCR experiment results show that Integrin αv-blocking antibodies can inhibit LGMN-induced metabolic reprogramming in M1 macrophages, while Integrin β3-blocking antibodies cannot inhibit LGMN-induced metabolic reprogramming in M1 macrophages (Figure 5G,H).

### 2.6. sLGMN Promotes Gastric Cancer Resistance to Anti-PD-1 Immunotherapy

Recently, the critical role of macrophage M2 polarization in tumor resistance to anti-PD-1 therapy has attracted a lot of attention. To further determine whether forced sLGMN expression is associated with the resistance of gastric cancer to anti-PD-1 immunotherapy, ELISA was used to detect the expression levels of serum sLGMN in 30 GC patients treated with PD-1 monoclonal antibody therapy. Patients were divided into two groups based on the median value of serum levels of LGMN (sLGMN^high^ and sLGMN^low^). The results indicated that patients in the sLGMN^high^ group had significantly shorter progression-free survival (PFS) compared to those in the sLGMN^low^ group (Figure 6A). To ascertain whether sLGMN induces resistance to PD-1 monoclonal antibody therapy in gastric cancer, we established a 615 mouse gastric cancer xenograft model using the MFC cell line. Recombinant mouse LGMN was subcutaneously injected to simulate high sLGMN expression. When the tumor volume reached approximately 100 mm^3^, intraperitoneal injections of anti-mouse PD-1 mAb were administered twice weekly for a total of six doses. After three weeks, tumor volume and weight were measured, and the inhibition rate for each group was calculated. The results showed that compared to the solvent control group, there was no significant difference in tumor growth in the LGMN group, suggesting that LGMN does not directly affect tumor growth in experimental animals. However, compared to the non-LGMN-treated group, tumors in the LGMN-treated group exhibited resistance to PD-1 mAb therapy (Figure 6B,C). These findings indicate that LGMN induces resistance to PD-1 mAb therapy in GC. These results indicate that sLGMN promotes GC resistance to anti-PD-1 immunotherapy in an M2-macrophage-dependent manner.

## 3. Discussion

Increasing evidence suggests that abnormal expression of LGMN is involved in various pathological processes [19,20,21,22]. Additionally, LGMN can be expressed in different locations, such as intracellularly or secreted into the extracellular space and expressed in the serum, indicating potential differential functions based on its expression localization [10,23]. By analyzing the expression levels of LGMN in the serum of GC patients sensitive and resistant to PD-1 mAbs, we found that the expression level of LGMN in the serum of anti-PD-1-resistant GC patients was significantly higher than that in sensitive patients. Among GC patients receiving PD-1 mAb therapy, high serum LGMN expression was associated with shorter progression-free survival. These results suggest that sLGMN may play a crucial role in PD-1 mAb resistance in GC. 

The communication between tumor-infiltrating immune cells and tumor cells plays a crucial role in forming an immunosuppressive microenvironment and inducing resistance to immune therapy [24,25]. TAMs, as one of the most important immune cell populations in the tumor microenvironment (TME), play critical roles in the occurrence and progression of various solid malignancies, including GC [26]. TAMs can be divided into classically activated M1-type macrophages and alternatively activated M2-type macrophages. M1-type macrophages play a significant role in anti-tumor immunity by secreting pro-inflammatory cytokines, whereas M2-type macrophages promote tumor progression and immune evasion by releasing anti-inflammatory cytokines [27]. Recent studies have shown that TAMs induce CD8^+^ T cell exhaustion by expressing immune checkpoint molecules such as PD-L1, PD-L2 (Programmed death ligand 2), CD86, and CD80 [26]. Furthermore, TAMs inhibit the function of T cells and dendritic cells by releasing IL-10, TGF-β (transforming growth factor-β), and PGE2 (Prostaglandin E2) [26,28]. Targeting TAMs is considered an effective approach to enhance the efficacy of immune checkpoint inhibitors due to their immunosuppressive properties. In GC tissues, TAMs inhibit the infiltration of CD8^+^ T cells by secreting CXCL8 (Interleukin-8), which also induces the expression of PD-L1 on macrophages to suppress the function of CD8^+^ T cells [29]. Here, we analyzed TAMs in the tumor tissues of GC patients treated with PD-1 mAbs. The results indicate that there is a higher abundance of CD206^+^ M2-type macrophage infiltration in the tumor tissues resistant to PD-1 mAbs. Building upon previous research, our study reveals a novel interaction network between GC cells and tumor-infiltrating macrophages that GC cells induce metabolic reprogramming on macrophages by secreting LGMN externally, promoting macrophage polarization towards the M2 phenotype, thus reshaping the GC immune microenvironment and ultimately forming a tumor environment resistant to PD-1 mAbs. 

Integrins are a class of heterodimeric surface receptors expressed on the cell membrane, involved in the interaction between tumor cells and arterial vessels [30]. Expression of Integrin αv and LGMN is significantly elevated in malignant tumor cells and the vascular system [16,30]. Integrins are expressed in fibroblasts, mesenchymal cells, tumor cells, and immune cells, and can cascade-activate downstream signals controlling cell proliferation, development, and adhesion. LGMN is typically located in lysosomes and mediates the processing of various albumins, such as converting cysteine cathepsin from a single-stranded to a double-stranded form [16]. Its high expression promotes tumor invasion, proliferation, and angiogenesis through multiple mechanisms. LGMN also shows high expression in TAMs, and the deletion of LGMN in TAMs can significantly accelerate tumor cell aging; however, the specific mechanisms are not yet clear [16]. For example, LGMN derived from breast cancer cells binds to Integrin αvβ3 on the cell surface through the RGD (Arginyl-glycyl-aspartic acid) motif, activating the FAK (Focal Adhesion Kinase)/Src (Steroid receptor coactivator)/RhoA (Ras homolog gene family member A) signaling pathway and promoting cancer cell migration and invasion independent of LGMN protease activity [10]. LGMN derived from macrophages binds to Integrin αvβ3 in VSMCs (vascular smooth muscle cells) and blocks Integrin αvβ3, thereby attenuating Rho-GTPase (Rho guanosine triphosphatases) activation, downregulating VSMC differentiation markers, and ultimately exacerbating TAD (thoracic aortic dissection) development [31]. Consistent with previous studies, our research found that secreted LGMN can bind to Integrin αvβ3 on the surface of M1-type macrophages, cascade-activating the PI3K (Phosphatidylinositol-4,5-bisphosphate3-kinase)/AKT/mTORC2 axis to regulate metabolic reprogramming and M2-type polarization, thus mediating GC resistance to PD-1 mAbs.

In this study, we first revealed the involvement of LGMN in GC resistance to PD-1 mAb therapy and elucidated its new mechanism by inducing metabolic reprogramming in M1-type macrophages and promoting M2-type polarization, ultimately leading to GC resistance to anti-PD-1 therapy. Our study provides a new potential therapeutic target and theoretical basis for reversing GC resistance to PD-1 mAbs and suggests the possibility of predicting the likelihood of GC patients becoming resistant to PD-1 mAb therapy based on serum LGMN expression levels.

## 4. Materials and Methods

### 4.1. Cell Lines and Clinical Tissues

The consecutive data of patients with locally advanced gastric cancer undergoing neoadjuvant chemotherapy and radical surgery were obtained from the database of the Department of Medical Oncology, Zhongshan Hospital, Fudan University. TCGA database (https://www.cancer.gov/tcga, accessed on 12 June 2023) is an unrestricted public database that can be directly applied globally without an ethical license. THP1 and GC cell lines were purchased from the cell bank of Shanghai Institutes for Biological Sciences, Chinese Academy of Sciences, and cultured using RPMI-1640 culture medium containing 10% FBS (Gbico, A5669701, New York, NY, USA) at 37 °C with 5% CO_2_. Our research was approved by the Ethics Committee of Zhongshan Hospital, Fudan University.

### 4.2. Immunohistochemistry

Immunohistochemistry was performed as described in our previous study [32]. In brief, tissue sections were incubated with primary antibodies against CD86 (Abcam, ab269587, Cambridge, UK) and CD206 (Abcam, ab8918, Cambridge, UK). For evaluation of the infiltration of M1 and M2 macrophages in the cancer tissues, the positive cells were counted under 4 areas of 200× images. The average positive cells were counted as the final score.

### 4.3. Transfection Experiment

The LGMN-overexpression lentiviral vectors were purchased from Genomeditech (Shanghai, China), and the blank lentiviral vectors were used as negative controls. The whole procedure of the transfection experiment was performed according to the manufacturer’s protocol. The Integrin β3 shRNA target sequences used in this study are listed as follows: shRNA-1: 3′-GGAUGAUCUGUGGAGCAUC-5′; shRNA-2: UGAUGCAUCCCACUUGCUG; shRNA-3: CUAUAGUGAGCUCAUCCCA.

### 4.4. Western Blot

The total proteins were extracted using a protein extraction kit according to the manufacturer’s instructions (#WLA019, Alibi, Hong Kong, China). Proteins were separated by 6% SDS-PAGE gel electrophoresis and then transferred to a PVDF membrane. The membranes were closed in 5% skimmed milk powder in Tris-buffered saline (pH 7.5) for 1 h at 37 °C. The membranes were incubated with the anti-human LGMN antibody overnight at 4 °C, followed by incubation with horseradish peroxidase-conjugated secondary antibody at room temperature for 1 h. Signals were detected using enhanced chemiluminescence reagent (Pierce, Rockford, IL, USA).

### 4.5. Enzyme-Linked Immunosorbent Assay

The concentrations of LGMN were measured by ELISA kits. The whole procedure of the ELISA experiment was performed following the manufacturer’s guidelines [33].

### 4.6. Extracellular Acidification Rate (ECAR) Analysis

ECAR analysis was performed with the Seahorse XF Glycolysis Stress Test Kit (103020-100; Agilent, Santa Clara, CA, USA) according to the manufacturer’s instructions as described in reference [34].

### 4.7. Flow Cytometry Analysis

Flow cytometry was performed to detect the infiltration of immune cells in subcutaneous tumors. The whole experimental procedure followed our previous study [32]. In addition, we strictly adhered to the standard operating procedures for flow cytometry, using appropriate isotype controls and fluorescence-minus-one (FMO) controls to ensure the accuracy and reliability of the results. Specifically, we included isotype controls for each fluorescent marker to rule out nonspecific binding effects. FMO controls were used to determine the negative boundary of each fluorescent signal, thereby enhancing the accuracy of our gating strategy. The final experimental results were analyzed using Flowjo (Version 10.8.1) software.

### 4.8. Lactic Acid Measurement

The level of lactic acid was detected using the CheKine™ Micro Lactate Assay Kit. The cells were collected into EP tubes after digestion with trypsin. We washed the cell samples with 1 mL of cold PBS, and then collected cells after centrifugation. Next, we used the lactate assay buffer to resuspend cells in a ratio of 1 mL/5,000,000, followed by ultrasonic crushing for 5 min (200 W, crush 3 s for every 7 s, repeat 30 times). Finally, the samples were centrifuged at 12,000× *g* for 5 min, and the supernatant was collected and detected at 450 nm.

### 4.9. Non-Targeted Metabolomics Sequencing Analysis

The non-targeted metabolomics sequencing analysis was conducted by Shanghai Bioprofile Co., Ltd. (Shanghai, China), using chromatography–mass spectrometry (LC-MS) technology. Metabolite structures were identified using precise mass number matching (mass tolerance < 20 ppm) and secondary spectrum matching (mass tolerance < 0.02 Da). Ion peaks (missing values > 50%) were deleted. Differential metabolites were analyzed using R studio (Version 4.1.1) according to the metabolite standard library.

### 4.10. ROS Production Assay

Gastric cancer cells were incubated with DCFH-DA (Beyotime, S0033S, Shanghai, China) diluted in 1:1000 serum-free culture medium at 37 °C for 20 min. Then, the cells were washed with the serum-free medium three times. The results were directly observed using a laser confocal microscope [35].

### 4.11. Animals

The 4-week-old male 615 normal mice were purchased from Nanjing Junke Biological Co. (Nanjing, China) MFC cells (1 × 106 cells/mouse) were subcutaneously injected into the right axillary posterior region. When the tumor volume reached approximately 100 mm^3^, the mice were divided into the following four groups: Recombinant mouse LGMN group; Recombinant mouse LGMN group + PD-1 mAb group; PD-1 mAb group; Solvent control group (PBS). PD-1 monoclonal antibody: 125 μg/mouse, intraperitoneal injection, twice a week. Recombinant mouse LGMN: 10 μg/mouse; subcutaneous injection, twice a week. A total of 6 treatments were administered. Tumor volume was measured every 3 days using the formula: length × width^2^ × 0.5. The animal experiments involved in this study were approved by the Ethics Committee of Zhongshan Hospital Affiliated with Fudan University.

### 4.12. Statistical Analysis

The results of continuous variables were presented as mean ± standard deviation (SD). Two groups of normally distributed and variance-aligned continuous variable data were analyzed using the *t*-test, multiple groups of normally distributed and variance-aligned continuous variable data were analyzed by one-way ANOVA, and data that did not satisfy parametric tests were analyzed by the Kruskal–Wallis method. Categorical variables were analyzed using chi-square tests or the Fisher exact probability method. Survival curves (Kaplan–Meier) were used to compare PFS between groups, and the differences were analyzed by Log-rank rows. GraphPad Prism (Version 9.0.0) and SPSS (Version 26.0.0) were used to analyze the data and draw graphs, and *p* < 0.05 indicated that the differences were statistically significant.

## Figures and Tables

**Figure 1 pharmaceuticals-17-00951-f001:**
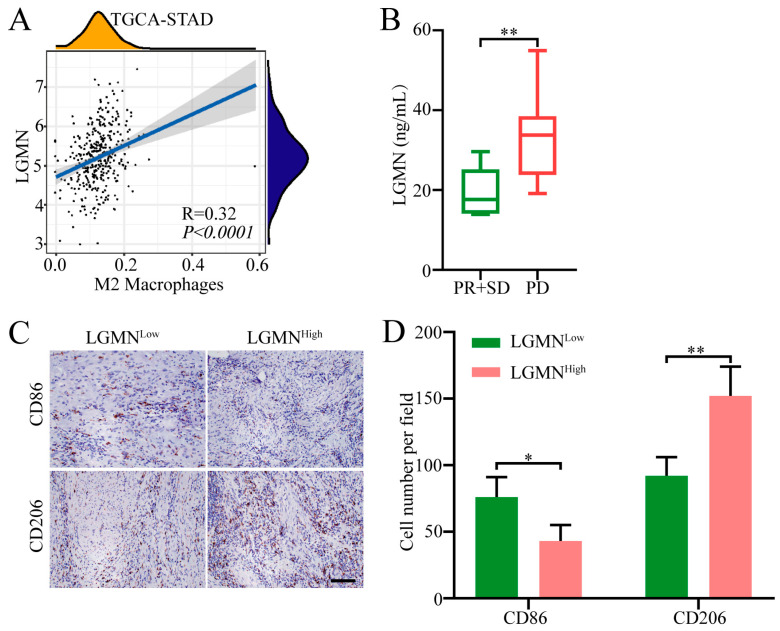
Relationship between LGMN expression levels and M2 macrophage infiltration in gastric cancer. (**A**) Analysis of TCGA database suggests a correlation between LGMN expression levels in gastric cancer and M2 macrophage polarization. (**B**) Differential serum LGMN expression in patients sensitive and resistant to anti-PD-1 therapy. (**C**,**D**) Immunohistochemical detection of M1 or M2 macrophage infiltration in gastric cancer tissues, and their correlation with serum LGMN expression levels in gastric cancer patients (bar 100 μm). * *p* < 0.05; ** *p* < 0.01.

**Figure 2 pharmaceuticals-17-00951-f002:**
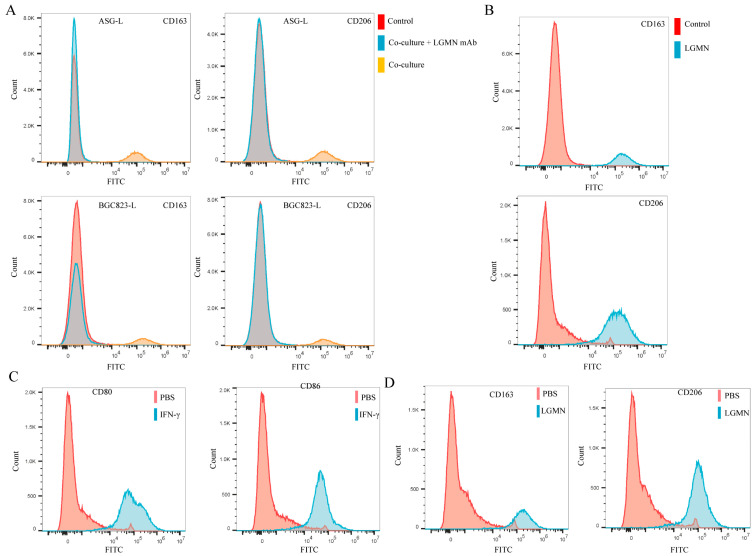
LGMN induces polarization of macrophages from M1 to M2 phenotype. (**A**) Co-culture of THP-1-derived M1 macrophages with ASG-L and BGC823-L cells followed by flow cytometry to detect expression of M2 macrophage markers CD163 and CD206, and the polarization of M1 macrophages towards M2 macrophages can be blocked by LGMN-neutralizing antibodies. (**B**) Incubation of THP-1-derived M1 macrophages with rh-LGMN protein for 5 days followed by flow cytometry to detect expression of M2 macrophage markers CD163 and CD206. (**C**) Stimulation of healthy human PBMCs with IFN-γ followed by flow cytometry to detect expression of CD80 and CD86. (**D**) Induction of M1 macrophages from PBMCs by culture with rh-LGMN, followed by flow cytometry to detect expression of CD206 and CD163.

**Figure 3 pharmaceuticals-17-00951-f003:**
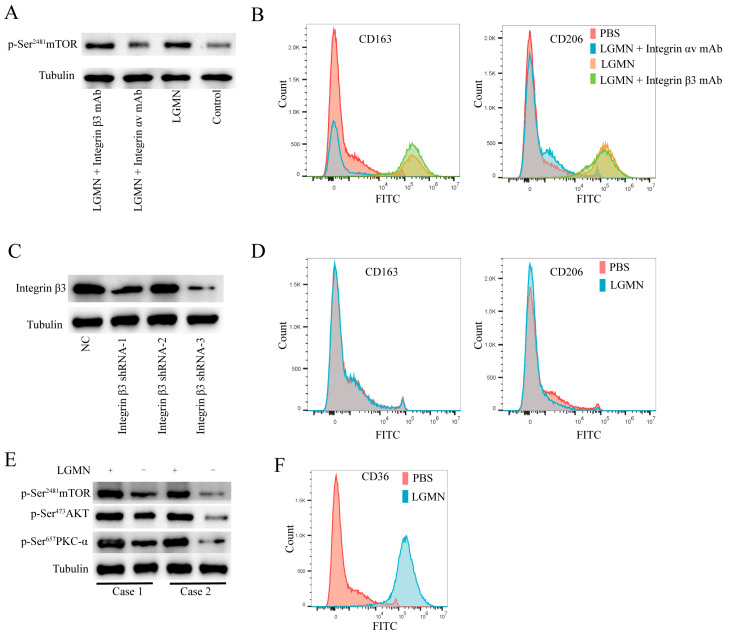
LGMN induces polarization of macrophages from M1 to M2 phenotype by forming a complex with Integrin αvβ3. (**A**) Western blot analysis of the effect of rh-LGMN, rh-LGMN + Integrin αv antibody, and rh-LGMN + Integrin β3 antibody on mTORC2 activity in M1 macrophages. (**B**) Flow cytometry analysis of the effect of rh-LGMN, rh-LGMN + Integrin αv antibody, and rh-LGMN + Integrin β3 antibody on the polarization of macrophages from M1 to M2 phenotype. (**C**) Establishment of M1 macrophage cells derived from THP-1 cells with knockdown of Integrin β3 using lentivirus. (**D**) Flow cytometry analysis of the effect of knocking down Integrin β3 expression on LGMN-mediated polarize macrophages from M1 to M2 phenotype. (**E**) Stimulation of M1 macrophages derived from PBMCs with rh-LGMN protein for 48 h followed by Western blot analysis to detect changes in mTORC2 signaling pathway activity. (**F**) Stimulation of M1 macrophages derived from healthy human PBMCs with rh-LGMN for 48 h followed by flow cytometry to detect changes in CD36 expression.

**Figure 4 pharmaceuticals-17-00951-f004:**
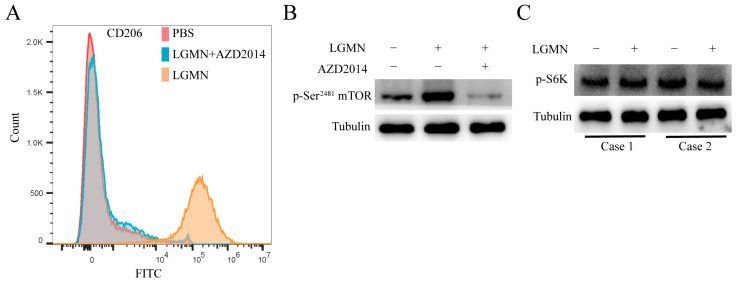
mTOR inhibitor blocks LGMN-induced polarization of macrophages from M1 to M2 phenotype. (**A**) Flow cytometry analysis of the effect of AZD2014 on LGMN-mediated polarization of macrophages from M1 to M2 phenotype. (**B**) Western blot analysis of the effect of AZD2014 on upregulation of mTORC2 signaling pathway activity by rh-LGMN in M1 macrophages. (**C**) Western blot analysis of the effect of rh-LGMN on mTORC1 signaling pathway activity in M1 macrophages.

**Figure 5 pharmaceuticals-17-00951-f005:**
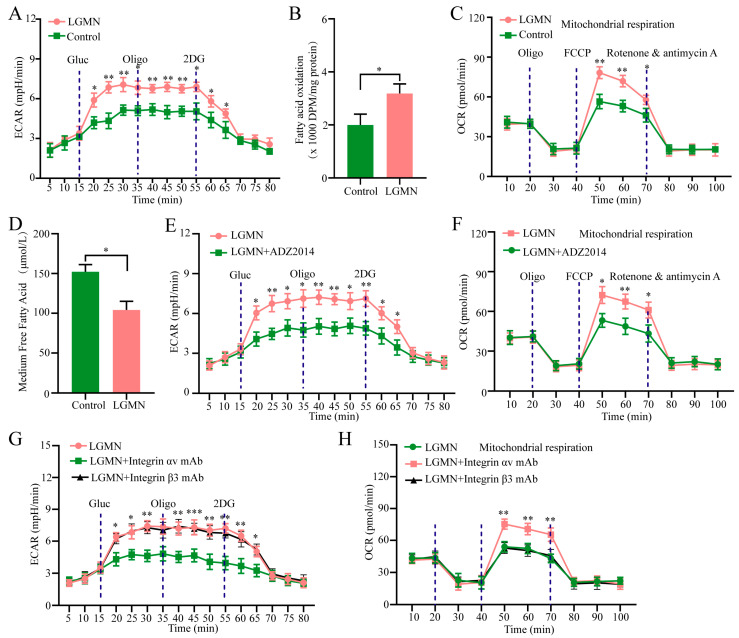
rh-LGMN protein promotes glycolysis and fatty acid oxidation in M1 macrophages derived from healthy human PBMCs. (**A**) ECAR assay. (**B**) Fatty acid oxidation assay. (**C**) Oxygen consumption assay. (**D**) Fatty acid uptake assay. (**E**,**F**) Low-dose mTOR inhibitor AZD2014 inhibits rh-LGMN-induced glycolysis and oxidative phosphorylation in M1 macrophages. (**G**,**H**) Blocking antibodies against Integrin αv but not Integrin β3 inhibits rh-LGMN protein-induced glycolysis (**G**) and oxygen consumption (**H**) in M1 macrophages derived from healthy human PBMCs. * *p* < 0.05; ** *p* < 0.01; *** *p* < 0.001 ((**G**,**H**): LGMN vs. LGMN + Integrin αv).

**Figure 6 pharmaceuticals-17-00951-f006:**
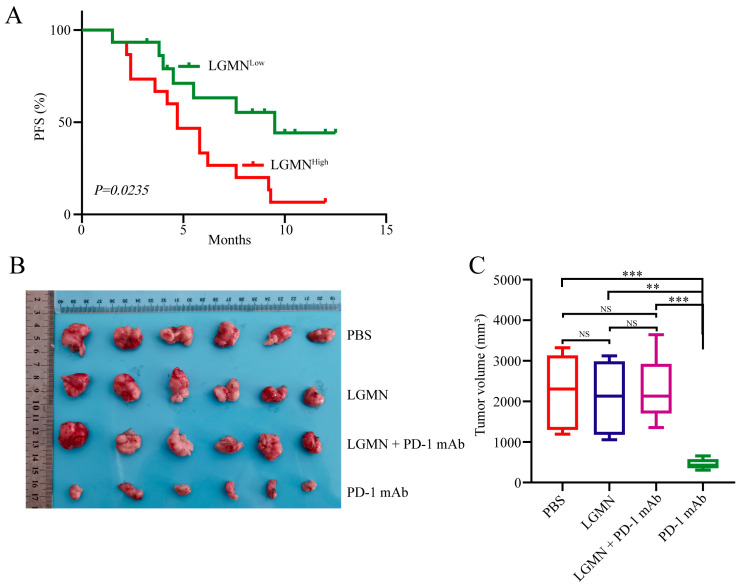
sLGMN induces resistance of gastric cancer to anti-PD-1 therapy. (**A**) Relationship between serum LGMN levels and PFS of gastric cancer to anti-PD-1 therapy. (**B**) Establishment of 615 mouse gastric cancer model using MCF cells, followed by different therapies to observe the relationship between sLGMN and sensitivity of gastric cancer to PD-1 mAb. (**C**) Statistical analysis of tumor volume after different treatments in 615 mouse MFC cell gastric cancer model. ** *p* < 0.01; *** *p* < 0.001; NS, not significant.

## Data Availability

Data are contained within the article.

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
