# Peer review of "Cancer Cell Secreted Legumain Promotes Gastric Cancer Resistance to Anti-PD-1 Immunotherapy by Enhancing Macrophage M2 Polarization"

_pharmaceuticals, 2024, doi:10.3390/ph17070951_

Round 1
Reviewer 1 Report
Comments and Suggestions for Authors
In the manuscript entitled “Cancer cell secreted legumain promotes gastric cancer resistance to anti-PD-1 immunotherapy by enhancing macrophage M2 polarization” the authors describe the investigation of a novel resistance pathway involving secreted legumain to anti-PD-1 immunotherapy in gastric cancer. Below are the comments to improve the manuscript.
1. When the GC secreted sLGMN interacts with surface integrin receptors of TAMs, what happens to the expression of intracellular LGMN in TAMs? What is the relative contribution of intracellular LGMN versus sLGMN towards resistance mechanism/s in TAMs? The authors should at least discuss this in the manuscript.
2. Did the authors consider more markers for the determination of M2 macrophages (TGFβ, IL-10, PGE2, CXCL8, Arginase)?
3. Did the authors use appropriate isotype and fluorescence minus one (FMO) controls for flow cytometry? What is the gating strategy? These details should be described in detail.
4. The authors should display densidometric quantitation of Westerns?
5. Lines 72-73: “In this study, our results demonstrated that GC cells exhibit elevated expression and secretion of LGMN. Subsequently, through its binding to integrin αv, LGMN upregulates…”
The authors should check and correct such mistakes throughout the manuscript.
6. Lines 92-93: “These results demonstrated that LGMN overexpression might be a promising molecular target for predicting the efficacy of immune therapeutics of gastric cancer”.
7. Abbreviations should be expanded the first time. The positive sign (+) should be denoted as a superscript.
Comments on the Quality of English LanguageThe authors should perform moderate English language edits of the entire manuscript.
Author Response
Reviewer 1
In the manuscript entitled “Cancer cell secreted legumain promotes gastric cancer resistance to anti-PD-1 immunotherapy by enhancing macrophage M2 polarization” the authors describe the investigation of a novel resistance pathway involving secreted legumain to anti-PD-1 immunotherapy in gastric cancer. Below are the comments to improve the manuscript.
- When the GC secreted sLGMN interacts with surface integrin receptors of TAMs, what happens to the expression of intracellular LGMN in TAMs? What is the relative contribution of intracellular LGMN versus sLGMN towards resistance mechanism/s in TAMs? The authors should at least discuss this in the manuscript.
Response: Thank you very much for your helpful comments. We discussed the aforementioned issues and added them to the revised manuscript.
- Did the authors consider more markers for the determination of M2 macrophages (TGFβ, IL-10, PGE2, CXCL8, Arginase)?
Response: Thank you very much for your helpful comments. CD206 combined with CD163 are currently widely recognized molecular markers of M2 macrophages. The present research findings support our conclusion; therefore, we did not further investigate other molecular markers.
- Did the authors use appropriate isotype and fluorescence minus one (FMO) controls for flow cytometry? What is the gating strategy? These details should be described in detail.
Response: Thank you very much for your helpful comments. In flow cytometry analysis, we strictly adhered to the standard operating procedures for flow cytometry, using appropriate isotype controls and fluorescence-minus-one (FMO) controls to ensure the accuracy and reliability of the results. Specifically, we included isotype controls for each fluorescent marker to rule out nonspecific binding effects. FMO controls were used to determine the negative boundary of each fluorescent signal, thereby enhancing the accuracy of our gating strategy. We have added above content on methods section in the revised manuscript.
- The authors should display densidometric quantitation of Westerns?
Response: Thank you very much for your suggestion. Our western blot results show significant differences between the groups, and expression differences can be observed even without quantitative analysis. We referred to several influential studies that also did not perform grayscale quantification for western blot results. Therefore, we did not quantify these results (References: Nature. 2023;624(7990):154-163. Mol Cancer. 2024 Apr 20;23(1):78. et al).
- Lines 72-73: “In this study, our results demonstrated that GC cells exhibit elevated expression and secretion of LGMN. Subsequently, through its binding to integrin αv, LGMN upregulates…” The authors should check and correct such mistakes throughout the manuscript.
Response: Thank you very much for your helpful comments. In the revised manuscript, we have corrected the inappropriate descriptions.
- Lines 92-93: “These results demonstrated that LGMN overexpression might be a promising molecular target for predicting the efficacy of immune therapeutics of gastric cancer”.
Response: Thank you very much for your helpful comments. In the revised manuscript, we have corrected the inappropriate descriptions.
- Abbreviations should be expanded the first time. The positive sign (+) should be denoted as a superscript.
Response: Thank you very much for your helpful comments. In the revised draft, the modifications have been made according to your suggestions.
Reviewer 2 Report
Comments and Suggestions for Authors
Review of the MS ID pharmaceuticals-3053385 titled “Cancer cell secreted legumain promotes gastric cancer resistance to anti-PD-1 immunotherapy by enhancing macrophage M2 polarization”
The paper explored the role of cancer cell-secreted legumain (LGMN) in influencing the tumor immune microenvironment and facilitating resistance to anti-PD-1 immunotherapy in gastric cancer.
The study utilizes a blend of in vitro cell-based tests, in vivo animal models, patient sample analysis, and molecular biology tools to clarify the underlying mechanisms.
just minimal typos/errors noted.
In introduction:
Lines 72-82 should be in conclusion. I suggest adding a paragraph formulating the problematic and the experimental design.
Here are a few instances of minor typographical errors:
In the introduction, the phrase "specifically cleaving peptide bonds within asparaginyl residues in the mammalian genome" should be revised to "specifically cleaving peptide bonds with asparaginyl residues".
The phrase "LGMN overexpression might be a promising molecular for predict the immune therapeutic effective of gastric cancer" should be revised to "LGMN overexpression might serve as a promising molecular marker for predicting the efficacy of immune therapeutic interventions in gastric cancer."
Use consistent formatting and styling for the figures and legends throughout the paper. This includes font size, line weights, and color schemes. Consistent formatting can help improve the overall visual appeal of the paper and make it easier to read.
Ensure that the figures are of high resolution and are legible when printed. This includes ensuring that text and labels are large enough to read easily and that images are not pixelated or blurry.
Author Response
Reviewer 2
Review of the MS ID pharmaceuticals-3053385 titled “Cancer cell secreted legumain promotes gastric cancer resistance to anti-PD-1 immunotherapy by enhancing macrophage M2 polarization”
The paper explored the role of cancer cell-secreted legumain (LGMN) in influencing the tumor immune microenvironment and facilitating resistance to anti-PD-1 immunotherapy in gastric cancer.
The study utilizes a blend of in vitro cell-based tests, in vivo animal models, patient sample analysis, and molecular biology tools to clarify the underlying mechanisms.
just minimal typos/errors noted.
In introduction:
Lines 72-82 should be in conclusion. I suggest adding a paragraph formulating the problematic and the experimental design.
Response: Thank you very much for your helpful comments. In the revised manuscript, we have made revisions based on your suggestions.
Here are a few instances of minor typographical errors:
In the introduction, the phrase "specifically cleaving peptide bonds within asparaginyl residues in the mammalian genome" should be revised to "specifically cleaving peptide bonds with asparaginyl residues".
The phrase "LGMN overexpression might be a promising molecular for predict the immune therapeutic effective of gastric cancer" should be revised to "LGMN overexpression might serve as a promising molecular marker for predicting the efficacy of immune therapeutic interventions in gastric cancer."
Response: Thank you very much for your helpful comments. We apologize for the errors caused by our lack of attention. We have made corrections in the revised manuscript.
Use consistent formatting and styling for the figures and legends throughout the paper. This includes font size, line weights, and color schemes. Consistent formatting can help improve the overall visual appeal of the paper and make it easier to read. Ensure that the figures are of high resolution and are legible when printed. This includes ensuring that text and labels are large enough to read easily and that images are not pixelated or blurry.
Response: Thank you very much for your helpful comments. In the manuscript, the format is written according to the template provided by the editorial office.
Reviewer 3 Report
Comments and Suggestions for Authors
Pei and colleagues investigated whether cancer cell secreted legumain could promotes gastric cancer resistance to anti-PD-1 immunotherapy by enhancing macrophage M2 polarization.
By in silico analysis and by analysing 20 patients samples the authors found that forced expression of LGMN in gastric cancer serum correlates with increased M2 macrophages infiltration in GC tissues and predicted resistance to anti-PD-1 immunotherapy. They used different approaches (cellular, molecular, biochemical assays) to investigate the mechanism of action of LGMN. They found that, gastric cancer cells secreted LGMN via binding to cell surface integrin αvβ3, then, activates Integrin 29 αvβ3/PI3K/AKT/mTORC2 signaling, thus promoting M1 polarization into M2 phenotype. By abolishing LGMN/Integrin αvβ3 interaction they observed inhibition of metabolic reprogramming and polarization of macrophages from M1 to M2 phenotype.
the work is well written, the results are interesting and the figures are representative of the results. The job can be accepted after clarification. In the flow cytometric analysis of co-cultures it is not clear how the macrophages were "separated" from the tumor cells for the purposes of marker analysis
Author Response
Reviewer 3
Pei and colleagues investigated whether cancer cell secreted legumain could promotes gastric cancer resistance to anti-PD-1 immunotherapy by enhancing macrophage M2 polarization.
By in silico analysis and by analysing 20 patients samples the authors found that forced expression of LGMN in gastric cancer serum correlates with increased M2 macrophages infiltration in GC tissues and predicted resistance to anti-PD-1 immunotherapy. They used different approaches (cellular, molecular, biochemical assays) to investigate the mechanism of action of LGMN. They found that, gastric cancer cells secreted LGMN via binding to cell surface integrin αvβ3, then, activates Integrin 29 αvβ3/PI3K/AKT/mTORC2 signaling, thus promoting M1 polarization into M2 phenotype. By abolishing LGMN/Integrin αvβ3 interaction they observed inhibition of metabolic reprogramming and polarization of macrophages from M1 to M2 phenotype. the work is well written, the results are interesting and the figures are representative of the results. The job can be accepted after clarification. In the flow cytometric analysis of co-cultures it is not clear how the macrophages were "separated" from the tumor cells for the purposes of marker analysis.
Response: Thank you very much for your helpful comments. For the co-culture, a chamber that does not allow cell passage was used for stratified cultivation. You may refer to our research group's previously published papers for more details (J Hematol Oncol. 2021 Nov 27;14(1):200.).
Reviewer 4 Report
Comments and Suggestions for Authors
In this study, Pei et al. found that s excreted legumian in gastric cancer serum correlates with increased M2 macrophages infiltration in GC tissues and predicted resistance to anti-PD-1 immunotherapy. Blocking legumain significantly inhibits metabolic reprogramming and polarization, which is a potent therapeutic target. Although interesting, while additional clarification and data are encouraged to supply.
1. Legumain can play a critical role in promoting M1-to-M2 polarization of THP-1-derived M1 macrophages, as shown in Figure 2A and B. However, the role of legumain is dominant since neutralizing legumain with antibody dramatically block M1-to-M2 polarization. That is dramatic, which means other factors may not affect macrophage polarization. Please clarify.
2. The authors should evaluate that whether inhibiting legumain expression in vivo can reverse the polarization of tumor-associated macrophages.
3. Also, for the anti-GC study, if the authors want to claim that legumain is a major reason contributing to resistance to anti-PD-1 therapy. They should not inject recombinant mouse legumain given that GC are overexpressing legumain. What they should evaluate is to knocking down or knocking out legumain to see if therapeutic response to anti-PD-1 therapy is increased.
Author Response
Reviewer 4
In this study, Pei et al. found that s excreted legumian in gastric cancer serum correlates with increased M2 macrophages infiltration in GC tissues and predicted resistance to anti-PD-1 immunotherapy. Blocking legumain significantly inhibits metabolic reprogramming and polarization, which is a potent therapeutic target. Although interesting, while additional clarification and data are encouraged to supply.
- Legumain can play a critical role in promoting M1-to-M2 polarization of THP-1-derived M1 macrophages, as shown in Figure 2A and B. However, the role of legumain is dominant since neutralizing legumain with antibody dramatically block M1-to-M2 polarization. That is dramatic, which means other factors may not affect macrophage polarization. Please clarify.
Response: Thank you very much for your helpful comments. The molecular mechanisms by which gastric cancer induces macrophage polarization to the M2 type are complex, involving factors such as exosome-mediated intercellular communication and abnormal expression of circRNAs et al. In this study, we investigated by exogenously upregulating LGMN in the AGS and BGC823 cell lines. Therefore, in the models used in this study, LGMN plays a dominant role in promoting the polarization of macrophages from the M1 type to the M2 type. While we do not exclude other potential molecular mechanisms, our experimental results show that blocking LGMN can prevent the macrophage polarization from M1 to M2 induced by the co-culture system. We speculate that the expression levels of other molecules promoting M1 to M2 macrophage polarization are insufficient in the selected cell lines to cause significant changes in M1 macrophages.
- The authors should evaluate that whether inhibiting legumain expression in vivo can reverse the polarization of tumor-associated macrophages. Also, for the anti-GC study, if the authors want to claim that legumain is a major reason contributing to resistance to anti-PD-1 therapy. They should not inject recombinant mouse legumain given that GC are overexpressing legumain. What they should evaluate is to knocking down or knocking out legumain to see if therapeutic response to anti-PD-1 therapy is increased.
Response: We greatly appreciate the valuable comments and suggestions from the reviewers. In this study, we discovered that high expression of LGMN in tumor cells is associated with PD-1 resistance in gastric cancer. Our investigation into the underlying mechanisms revealed that mTORC2 is a key downstream molecule of sLGMN. Currently, mTOR inhibitors have entered the clinical research stage. Given the accessibility of these experimental drugs and the need for subsequent translational research, we chose to study mTOR inhibitors. Our findings will better facilitate the guidance and implementation of future clinical application studies.
Reviewer 5 Report
Comments and Suggestions for Authors
1.These results indicate that sLGMN promotes GC resistance to anti-PD-1 immunotherapy in a M2 macrophages dependent manner (Fig. 6B and C).
If you want to get the conclusion, one more group of mouse experiment should be included: sLGMN+anti-PD-1+macrophage depletion. In addition, survival curve should be shown. Of note, the author tell us that sLGMN is one of important factors for anti-PD-1 therapy failure. Why you choose anti-PD-1 therapy sensitive tumor cell line? Actually, you should choose the tumor cell line with two characteristics: expressing high level of sLGMN and resisting to PD-1 antibody theray.
2.The author tell us that sLGMN derived from tumor cells promotes M1 to M2 polarization. So you should use LGMN knock down tumor cells to perform the animal experiments,and demonstrat that secreted LGMN in tumor tissues is downregulated, and the number of M1 and M2 macrophages is changed.
3.In Fig. 1C, the tumor samples from patients should be examined for LGMN expression, and analyze if the LGMN expression in tumor cells are positively related with serum LGMN level.
4.In Fig. 2A, LGMN over-expression tumor cell lines are constructed. To demonstrate your conclusion, you should choose tumor cell lines expressing high level of LGMN, and then knock down its expression, and assess its influence on macrophage polarization.
Comments on the Quality of English LanguageMinor editing of English language is required.
Author Response
Reviewer 5
1.These results indicate that sLGMN promotes GC resistance to anti-PD-1 immunotherapy in a M2 macrophages dependent manner (Fig. 6B and C). If you want to get the conclusion, one more group of mouse experiment should be included: sLGMN+anti-PD-1+macrophage depletion. In addition, survival curve should be shown. Of note, the author tell us that sLGMN is one of important factors for anti-PD-1 therapy failure. Why you choose anti-PD-1 therapy sensitive tumor cell line? Actually, you should choose the tumor cell line with two characteristics: expressing high level of sLGMN and resisting to PD-1 antibody theray.
Response: We greatly appreciate the valuable comments and suggestions from the reviewers. In this study, clinical data indicated that high expression of LGMN in tumor cells is associated with PD-1 resistance in gastric cancer. Our investigation into the underlying mechanisms revealed that mTORC2 is a key downstream molecule of sLGMN. Currently, mTOR inhibitors have entered the clinical research stage. Given the accessibility of these experimental drugs and the need for subsequent translational research, we chose to study mTOR inhibitors. Our findings will better facilitate the guidance and implementation of future clinical application studies.
Additionally, the tumor models established with two types of cells better support our research conclusions. However, immune reconstitution poses some challenges in our current situation, and the only available murine gastric cancer cell line is MFC. Therefore, we only selected one animal model, which is a limitation of our study. In future research, we will strive to create better conditions for further investigation.
- The author tell us that sLGMN derived from tumor cells promotes M1 to M2 polarization. So you should use LGMN knock down tumor cells to perform the animal experiments,and demonstrat that secreted LGMN in tumor tissues is downregulated, and the number of M1 and M2 macrophages is changed.
Response: Thank you very much for your helpful comments. Our experimental results have confirmed that LGMN overexpression can promote the polarization of M1 macrophages to the M2 type. These findings strongly support our research conclusions. LGMN can be localized within tumor cells and secreted outside the cells, with different biological functions depending on its location. In this study, we primarily investigated the biological function of sLGMN, using recombinant LGMN to simulate its high expression. Additionally, the molecular mechanisms by which tumor cells induce macrophage polarization from M1 to M2 are complex and involve multiple factors, such as exosome-mediated intercellular communication and paracrine signaling. Knocking down LGMN expression in tumor cells might still lead to M1 to M2 macrophage polarization through other mechanisms. Therefore, we did not pursue further knockdown studies.
3.In Fig. 1C, the tumor samples from patients should be examined for LGMN expression, and analyze if the LGMN expression in tumor cells are positively related with serum LGMN level.
Response: Thank you very much for your helpful comments. Based on the reviewers' comments, we detected the expression levels of LGMN mRNA in tumor tissues from 20 patients using qPCR. The results showed that the expression level of LGMN mRNA in tumor tissues was positively correlated with the expression level of LGMN protein in serum (Figure 1). Furthermore, in Supplementary Figure 1, we upregulated LGMN expression in gastric cancer cells using lentivirus, which led to increased LGMN expression in both the gastric cancer cells and the culture supernatant. These findings support that the increased expression of LGMN in gastric cancer cells is accompanied by an increase in LGMN secretion.
Figure 1. the expression level of LGMN mRNA in tumor tissues was positively correlated with the expression level of LGMN protein in serum
- In Fig. 2A, LGMN over-expression tumor cell lines are constructed. To demonstrate your conclusion, you should choose tumor cell lines expressing high level of LGMN, and then knock down its expression, and assess its influence on macrophage polarization.
Response: Thank you very much for your helpful comments. Our experimental results have confirmed that LGMN overexpression can promote the polarization of M1 macrophages to the M2 type. These findings effectively support our research conclusions. Additionally, the molecular mechanisms by which tumor cells induce macrophage polarization from M1 to M2 are complex and involve multiple factors, such as exosome-mediated intercellular communication and paracrine signaling. Knocking down LGMN expression in tumor cells might still lead to M1 to M2 macrophage polarization through other mechanisms. Therefore, we did not pursue further knockdown studies.

Round 2
Reviewer 5 Report
Comments and Suggestions for Authors
The authors should resolve my concerns by doing experiments.